# Strategies to Enhance Knowledge and Practical Skills of Triage amongst Nurses Working in the Emergency Departments of Rural Hospitals in South Africa

**DOI:** 10.3390/ijerph18094471

**Published:** 2021-04-23

**Authors:** Thabo Arthur. Phukubye, Masenyani Oupa. Mbombi, Tebogo Maria. Mothiba

**Affiliations:** 1Department of Nursing Science, University of Limpopo, Mankweng 0727, South Africa; masenyani.mbombi@ul.ac.za; 2Faculty of Health Science, University of Limpopo, Mankweng 0727, South Africa; tebogo.mothiba@ul.ac.za

**Keywords:** triage, strategies, continuous training, knowledge and skills, practice, emergency departments

## Abstract

*Purpose:* Lack of knowledge and practical skills on triage remains a global problem, especially within rural hospitals, and very little is known about enhancing the knowledge and skills of emergency nursing staff in rural hospitals of South Africa. The objective was to describe the perceived strategies for enhancing knowledge and practices of triage among nurses working in the emergency departments (EDs) of rural hospitals in South Africa. *Research methods:* A descriptive qualitative research design was applied to achieve the research objective. A non-probability sampling method was applied to select 17 professional nurses from rural hospitals. An unstructured face-to-face interview method was used to collect data. Data collected were analyzed using Tesch’s method of data analysis. *Results:* The study findings highlighted the academic needs of nurses working in the ED of rural hospitals. Two themes emerged from this study; (1) The consistent description of the importance of triage training for emergency unit staff, and (2) The description of measures to enhance triage practices amongst emergency unit staff. Findings indicated that triage knowledge and practice remains a challenge, but with formulated strategies like continuous training by workshops, refresher courses, and offering a training module on triage, evaluation of developed guidelines and benchmarks is often enhanced. *Conclusions:* The study describes the strategies to enhance the conversion of knowledge and practice of triage amongst nurses working in the ED of rural hospitals. The paper argues that the knowledge and practical skills of nurses working in ED are enhanced through the provision of continuous training as workshops, triage module, evaluating the developed guidelines to implement triage, and benchmarking with other hospitals.

## 1. Introduction and Background

Knowledge and skills amongst emergency nurses are critical factors for quality nursing care in the Emergency Department (ED). For this reason, Kerie et al., [1] believe that there is a strong positive relationship between triage knowledge and practice as well as emergency nursing care. However, according to Considine, Botti and Thomas, [2] having knowledge about triage alone does not always yield good triage practical skills because knowledge is factual, and acquisition of real knowledge alone is not necessarily associated with behaviour change in terms of practice and clinical decisions [2]. For instance, Fathoni, Sangchan, and Songwathana [3] noted that knowledge alone is not enough to yield accurate clinical findings. Therefore, Kenneth, Iserson, Tucson, John, and Moskop [4] confirm that triage’s effectiveness is embedded in the emergency staff members’ knowledge and skills. The development of triage decision-making skills and practices can be enhanced and addressed through the use of simulations, “thinking aloud’ techniques, reflection, and the decision rules of the experienced emergency nurses. Therefore, triage nurses should have a proper education and proficiency in emergency triage, decision making, and emergency nursing care especially considering that training on triage is an integral part of emergency nursing education [5].

Various authors have investigated and deliberated on the level of knowledge and skills of nurses working in the ED regarding triage in South Africa [6,7,8,9,10,11]. For instance, Phukubye, Mbombi, and Mothiba [6] reported nurses in the Sekhukhune District of Limpopo Province in South Africa (SA) had an average score on knowledge of triage with poor practical skills. Similarly, a study by Dulandas and Brysiewicz [7] reported a need for a support system in educational development that will improve the competency, knowledge, and practical skill level among most emergency nurses. Brysiewicz [11], on the other hand, reported that poor triage practice of emergency nurses results from a lack of knowledge amongst nurses in ED. Therefore, Cimona-Malua [12] concluded that the triage system is inefficient and needs improvement especially because it contributes to prolonging patients’ waiting time in the Sekhukhune District of Limpopo Province in SA.

Lack of knowledge and practical skills among emergency nurses in the ED is not a South African healthcare problem alone but a global problem that affects even developing countries. For example, Aloyce, Leshabari, and Brysiewics [8] in Tanzania reported that lack of knowledge and skills is a problem amongst nurses working in ED. On the other hand, a study conducted by Ali, Bernice, Ghani, Kussor, and Naz [13] on the assessment of triage knowledge on nurses in three teaching hospitals of Pakistan revealed that sixty-nine percent (69%) of nurses attained low scores. In Sudan, the study conducted by Muktar et al. [14] indicated that thirty-three percent (33%) of nurses were not knowledgeable about triage, fifty-two percent (52%) were unable to allocate an appropriate triage category, which is practice, and fifty-eight percent (58%) lacked knowledge of the waiting time in the emergency department. Generally, these studies show that there are still serious concerns in different countries over triage nurses’ knowledge and practice. Furthermore, the low percentages of emergency nurses’ knowledge regarding triage in the ED suggest a gap that can be enhanced through the emergency staff personnel’s capacity building. The authors argued that there should be a standardized training system to appropriately train nurses in the emergency department about the triage system and other emergency care systems. 

In this regard, Mulindwa [15] noted that adequate and extensive clinical knowledge and emergency nurses’ skills regarding triage provide optimal emergency care. In acknowledging the significance of knowledge and practical skills of triage among nurses in the ED, very few strategies have been done to enhance the knowledge. For instance, Rankin, Then and Atack [16] developed and tested a web course that provided a standardized and compelling educational experience to enhance emergency nurse’s triage accuracy. WHO (a) [17] guidelines indicate training as a component for improving emergency care. However, very little was deliberated regarding strategies to achieve educational needs through training. Furthermore, the Emergency Nurses Association [18] recommended a comprehensive orientation program to support preceptors to meet mentees’ total educational needs by building knowledge and skills.

Given the above discussion on the background of knowledge and practical skills on triage, the authors concluded a need to formulate educational strategies to close the existing gap on the nurses working in the EDs. In this regard, the authors believe such a strategy might be of significance in improving and sustaining the patient outcomes and quality emergency nursing care within rural hospitals of SA. Therefore, the authors’ interest in this paper is to describe the perceived strategies for enhancing the knowledge and practices of triage among nurses working in the emergency departments of rural hospitals in South Africa.

## 2. Research Materials and Methods

The COREQ Checklist assisted the authors in writing the manuscript (see the Appendix A). A descriptive qualitative research design was used to describe the strategies to enhance knowledge and practical triage skills amongst nurses in ED. A descriptive qualitative design was appropriate to describe the perceived strategies for enhancing knowledge and practices of triage among nurses working in the emergency departments of rural hospitals, which is the objective of this study [19]. Furthermore, most of the studies that explored the knowledge and practice of triage among nurses used a quantitative approach (for example, studies by Cimona-Malua and Naidoo [12,20]) to achieve the objective.

### 2.1. Study Site

The study was conducted in five municipalities of the Sekhukhune District, providing 24-hour emergency services to patients in rural areas. The five municipalities had a total of 84 emergency nurses, as outlined in Table 1 below.

### 2.2. Population and Sampling

According to the National Government of South Africa [21], Limpopo Province has five districts: Sekhukhune, Capricorn, Vhembe, Mopane, and Waterberg District. The districts are further divided into local municipalities. The study was conducted in five municipalities under the Sekhukhune District, which provides 24 h emergency services to patients in rural areas. The five municipalities had a total population of 84 emergency nurses. Only 17 emergency nurses (that included categories of registered/professional nurses and speciality trained nurses) working within the EDs of rural hospitals of Sekhukhune District in Limpopo Province were purposively selected in the study. The sample’s total number was guided by data saturation from the total population. The authors employed non-probability purposive and convenience sampling types to sample nurses with more than two years of experience working in ED.

### 2.3. Data Collection

Turfloop Research Ethics Committee (TREC) gave ethical approval for the study to be conducted (TREC/372/207: P.G.). Permission to conduct the study within the hospitals was granted by the Limpopo Department of Health. Sekhukhune District office gave consent to conduct the study within the hospitals. Respective Chief Executive Officers (CEOs) of the five hospitals granted permission to approach nurses working within the emergency departments. Deputy Nurse Managers and Operational Managers of the ED assisted in arranging dates and quiet areas for the data collection. Data collection was organized to happen during the day in the working place when most nurses were available on duty. At the time of data collection, the authors established a relationship with the participants to obtain informed consent. Furthermore, the authors gave instructions and the rationale for the research study to the participants as well as their interest in the research topic. The unstructured face-to-face interviews were conducted by one author only to collect data in a quiet and controlled environment. Emergency nurses were asked to describe their perceived strategies that can enhance the knowledge and practice of triage among nurses in the ED. Follow-up probing questions were then used for emergency nurses to give more explanation. Unstructured interviews were conducted until data saturation was reached and the interviews lasted between 15–25 min. Data were recorded on a digital audio recorder that was supplemented by taking field notes during interviews. The authors ensured the safety of digital audio recorded data with the use of an encrypted document with password access.

## 3. Analysis

Data collected were analyzed using Tesch’s coding method of data analysis, as recommended by [22]. Tesch’s coding method has proven to be very useful in qualitative data analysis [15,22]. The audio recorded data were transcribed verbatim by the first author who conducted the interviews to facilitate themes, sub-themes, and categories generation and identification. The first two authors read all the transcripts one by one to obtain the meaning of the data. All related data were coded into categories. The categories were formulated by combining all associated data as coded by the first two authors. The acquired categories were clustered together to generate potential themes and sub-themes. To ensure the study’s dependability, the third author acted as an independent coder during the data analysis process. The developed themes, sub-themes, and categories were verified and confirmed by all the authors only, however, participants were not informed about the feedback of data analysis.

## 4. Results and Discussion of Findings

Table 2 below describes the demographic data of professional nurses who participated in the study.

Table 2 indicates the study had a total of 17 emergency nurses. Of the 17 emergency nurses, nine were female, with the remaining eight being male professional nurses. The gender proportions indicate that females remain the dominant gender of the nursing profession, even though in this study’s results, the difference was not much [23]. The study had professional nurses with different qualifications, where nine professional nurses had a general qualification, one master’s trauma trained professional nurse, and seven diploma trauma trained professional nurses. The qualification proportions of emergency nurses still indicate a gap of trauma trained nurses in rural hospitals [24]. Table 2 shows most nurses were more experienced in emergency nursing and probably should be more knowledgeable and skilled in triage. 

Table 3 below provides a summary of the main findings according to the themes and sub-themes and categories.

## 5. Theme 1: The Consistent Description of the Importance of Triage Training for Emergency Unit Staff

The study findings indicated a consistent description of the importance of triage training for emergency unit staff. Emergency nurses reported that triage training could enhance the knowledge of triage amongst nurses in the emergency department. Participants emphasized the upskilling of staff on triage as an essential enhancement factor of the existing triage knowledge through various training platforms. The triage course/module as an orientation course and guidelines for implementing triage is further suggested as another strategy for enhancing triage knowledge among nurses in the emergency departments. The theme is discussed in detail according to the following sub-themes: 

### 5.1. Sub-Themes 1.1: Participants Emphasized the Upskilling of Emergency Nurses on Triage as an Important Enhancement Factor of the Existing Triage Knowledge through Various Training Platforms

Professional nurses indicated that knowledge and practice skills could be enhanced by continuous learning and development that can be offered using various platforms such as: in-service training, emergency drills, workshops, and on-going assessment. The participants indicated that the in-service training and workshops are critical programs in enhancing the knowledge of triage in the ED. Furthermore, participants argued that triage training should be a prerequisite for Emergency Unit staff while indicating the need for a refresher course that can be done annually. 

The illustration below supports the findings.

Participant 3 (<5 years of experience): “*By offering Training to all Emergency Department staff, and offer refresher courses annually*.”Participant 1 (>10 years of experience): “*Training and reinforcing information so that nurses don’t forget and evaluations, and correctional measures to help nurses know and be sure of what they are doing to be able to sort outpatients and save lives*.”Participant 16 (>10 years of experience): “*Providing training, re-triage, and assessment after each training using a questionnaire and providing emergency drills, delegate each one to triage, then discuss how it was after each case in Casualty will reinforce triage skills*.”Participant 12 (<less than % years of experience): “*On-going in-service training is key in enhancing the knowledge of triage*.”Participant 10 (<5 years of experience): “*Through continuous learning and development this includes in-service training, emergency drills, workshops, and on-going assessment (performance management systems)*.”

The study conducted by Rahmati, et al. [25], on the consequences of triage education on knowledge, practice, and qualitative index of ER staff indicate that triage nurses must have appropriate formal training and knowledge in emergency nursing triage, decision-making, and emergency care. Formal training in triage improves the effectiveness of triage nurses and, with improved confidence, nurses will be prepared to perform more efficiently. Rahmati et al. [25], on triage knowledge indicate that most nurses scored low marks on triage knowledge but that their marks improved after triage training. The authors above affirm that continued education on triage is vital to maintaining good and up so far triage skills. The study findings conducted by Fathoni et al. [3], provide a far better understanding of triage skills among Emergency Department nurses. They suggest that continuing education and training courses associated with triage and advanced management of medical emergencies for E.D. nurses are essential to extend and update triage skills to reinforce the standard of emergency care of patients. Thus, the more training or drills attended, the higher the talents that nurses develop, and therefore the better their triage knowledge and practice [3]. 

In contrast to current findings, Bruce et al. [26], reported simulation as the best strategy to enhance triage knowledge amongst emergency nurses as it provided nurses with an opportunity to practice. Clinical simulations can facilitate a learning process as they’re active and mimic reality [26]. Our findings are consistent with the study conducted by Rankin, Then, and Then [16] on the effectiveness of online triage learning; 74% of the respondents reported that simulations improved their triage skills. Simulations encourage deep understanding, help novice nurses to develop confidence, and enhance their clinical judgment.

### 5.2. Sub-Theme 1.2: Triage Course/Module as an Orientation Course and Guidelines for Implementation Triage Suggested

This study’s findings indicated that emergency nurses perceived that triage knowledge and practice could be enhanced by having a triage module for all nurses’ categories. The participants further reported that a yearly educational course for nurses in the ED is a vital step to take. The module will provide room for assessing and evaluating the learned triage skills at the end. The illustration below demonstrates these.

Participant 11 (>10 years of experience): “*Having a training module not only to enrolled nurse’s auxiliary but to all the ranks who deal with patient’s care. They need to familiarize themselves with the meaning of priority when you are doing triage. There should be workshops about triage situations*”.Participant 4 (Between 6–10 years of experience): “*There should be a triage module for a workshop with specifying objectives and strategies of triage for emphasis and evaluation of the triage skill should be conducted at the end of the workshop.*”Participant 8 (Between 6–10 years of experience): “*yearly educational course to all nurses in emergency departments about triage, particularly the new nurses in the emergency department, is vital*.”

The Emergency Nurses Association [18] in the United States of America stated that emergency medical care is a speciality care unit that requires a comprehensive orientation program to assist in quality patient care. We share an equivalent sentiment with the association by stating that having an orientation program may enhance nurses’ knowledge and skills while improving the competency level and readiness within the ED. This is often because the orientation program will provide access to the new staff members to an experienced triage nurse in the least time. Intrinsically, triage educational needs will be met. In this regard, the WHO (a) [17] guidelines on emergency care state that there should be standardized short courses for nurses working within the ED to reinforce their knowledge and practical skills on triage. Dulandas et al. [7], confirm that short courses in emergency care like Advanced Cardiac Life Support and Peads Life Support enhanced the knowledge and practical skills of nurses working within the E.D.

The use of recent technology in teaching and learning has proved to be effective, hence improving the health care systems. For instance, Rankin et al. [16], reported that triage knowledge and skills were enhanced using modern technology, like online learning, where a web-based course like the Canadian Triage and Acuity Scale (CTAS) course was found to be effective in enhancing practical skills of triage. Web learning can help nurses improve and maintain their competency, and it can support professional practice. The study’s findings indicate that 60% of respondents reported having enjoyed online triage learning, 74% had convenient computer access compared with 41% without computer access. Ninety-two percent thought group enrollment for online triage learning was an honest idea. The general results were that 78% of the respondents noted improved triage knowledge from web learning. Mandatory online tutorials, discussions, and workplace projects successfully transferred triage learning to practice [16]. In South Africa, web learning may be a possible strategy despite the challenges of infrastructure resources like computers and access to the web. Most nurses are young and use smartphones, which they will use to enroll in online triage courses to up their knowledge and practices. Although further research is required to supply more evidence of best teaching and learning practice in triage via web learning, nurses within the ED are often recommended to enroll for such training in rural hospitals.

## 6. Theme 2: Description of Measures to Enhance Practices amongst Emergency Unit Staff

The study findings indicated a description of measures to enhance practices amongst emergency unit staff. Professional nurses reported that a formulation of clear guidelines for handling patients in the triage area and benchmarking on managing the triage area as possible strategies to enhance triage practice. Furthermore, professional nurses expressed the important role that clinical supervisors can play in enhancing triage practice in the ED. The theme is discussed in detail according to the following sub-themes:

### 6.1. Sub-Theme 2.1: Formulation of Clear Guidelines for Handling Patients in the Triage Area Viewed as Important

Emergency nurses indicated that the practice of triage in ED could be enhanced by having guidelines and algorithms about triage available in the department. Furthermore, participants expressed that the implementation of these guidelines and algorithms should be quarterly audited for the correct implementation and evaluation of the effectiveness. Other professional nurses felt that making the information available everywhere in the unit on the triage colors’ interpretation about their urgency and waiting times might enhance the triage practice. These can be achieved by developing pamphlets about the triage service and procedures. The quotes below support the findings.

Participant 3 (<5 years of working experience): “*By having guidelines and algorithms available in the department, then have quarterly audits for the correct implementation of the guide, and evaluating the effectiveness*.”Participant 4 (>10 years of working experience): “*There should be information available to the public on the interpretation of triage colors about their urgency and waiting times*.”Participant 17 (>10 years of working experience): “*Having pamphlets on halls that elaborate different emergency care services, or rather procedures*.”

According to Diolaiuti, et al. [27], the algorithm has been studied in hospital Careggi (Tuscany, Italy), and 352 patients have been involved from October 2015 to March 2016. The results that came out were very significant. The accuracy was 88%, demonstrating high sensitivity (95%) and specificity (87%). With the algorithm in the ED, the waiting time for the patient was drastically reduced, and so were the costs made by the ED, avoiding a lot of imaging tests and other unnecessary specialist consults and ultimately leading to increasingly adequate management of resources. Given the variability in triage training and experience, there is a worldwide need to develop uniformly tailored triage education curricula and triage guidelines, as well as continuing training and research in triage systems, triage guidelines coupled with triage education and training helps triage nurses to prioritize ED patients in all health care settings including psychiatric EDs [28].

In a study conducted by Hategeka et al. [29], of implementing Emergency Triage Assessment Treatment (ETAT) and guidelines to improve Rwanda’s clinical practice, one participant was coded as follows:
*“I found hospital audit helpful and engaging, as I could see what was done wrong, and discussed with others what could have been done based on [ETAT+ clinical practice guidelines] recommendations.”*
*“Using the while booklet [Rwanda basic pediatrics protocols] as we were auditing medical records helped to integrate [ETAT+] materials.”*
*“The audit here [in ETAT+ training] seems friendly. We are learning not blaming.”*

The establishment of the triage audit committee has shown to be effective in enhancing the knowledge and skills of nurses regarding triage within the ED. As an example, Reinhart [30] reported that the formation of a panel of skilled nurses to assess and recommend potential solutions to the educational needs of nurses within the ED was a useful approach in Texas, USA. Furthermore, as reported in WHO (b) [31] about the triage audit committee, which was mandated to enhance nurses’ knowledge and practical skills, Qatar’s story achieved its mandate by developing a training program for meeting the tutorial needs of nurses within the emergency care unit. The committee further provided monitoring and evaluation regarding the knowledge and practical skills versus quality emergency care. The authors believe that hospitals in rural areas of SA might take the initiative of building committees that will be liable for enhancing nurses’ knowledge and practical skills working within ED.

### 6.2. Sub-Theme 2.2: Benchmarking on the Management of the Triage Area to Improve Practice Suggested

The findings indicated that triage practice could be enhanced when secondary hospitals conduct benchmarks in tertiary hospitals. Furthermore, professional nurses reported that triage practice could be enhanced by allowing nurses working in rural emergency hospitals to come and work in provincial while rotating between the EDs of the hospitals annually for exposure enhances on triage knowledge practice. The illustration below demonstrates these findings.

Participant 7 (>10 years of working experience): “*should also benchmark at other hospitals such as tertiary hospitals and see how it works*.”Participants 10 (<5 years of working experience): “*...including a...rotation of all nurses within the unit to the triage room*.”Participant 13 (Between 5–10 years of working experience): “*by allowing nurses who are working in rural emergency hospitals to come and work in the Provincial emergency department for exposure enhances triage knowledge and practice*.”Participant 16 (>10 years of working experience): “*Nurses can rotate from between the Casualty units of the hospitals annually; this will enhance their knowledge*.”

According to Benner’s theory [32] of skill acquisition, reflective practice is a tool that can be used to bridge the gap between theory and practice. This is often an important strategy because it will provide the hospital’s weaknesses and strengths. Reflective practice is often used to re-examine an experience to know and plan the way to act better during a similar situation within the future. In this regard, the reflective practice might assist the clinical supervisors of ED in planning a benchmark and a rotation plan between urban and rural hospitals. Beam [33] found that debriefing through reflective practice helps nurses manage stresses and emotions triggered by challenges, thus improving nursing practice. Reflection gives insight into practice; nurses can isolate areas of strength and areas that require further development within the Emergency Departments, like in triage, and provide a base for benchmarks and rotation purposes. Self-assessment and reflection allow nurses to think about their practice within their environment and help them sustain and increase their practice. Reflective practice enhances the nurse’s critical thinking and judgment supported experience and prior knowledge and eventually enhances patient care and practice. The study by Wolf et al. [34], shows that there’s little triage content presented as a part of undergraduate or postgraduate nursing programs, so rotation to emergency care centers, which are far better equipped and have proper protocols, will assist in enhancing nurses’ knowledge and assist them in practising effectively within the emergency department. We believe that counting on the allocated centers and the rotation process may enhance both the knowledge and practical skills of the newly employed nurses and those in the system for a while. Therefore, establishing collaboration of healthcare managers in urban and rural hospitals becomes a relevant and important factor for ensuring a sustainable rotation of nurses between the hospitals, especially since the current study was conducted mainly in rural hospital areas. Consistent with Rahmati et al. [25], it’s better to use experienced nurses within the ED for the triage of patients than inexperienced emergency nurses. The use of experienced nurses within the ED might help achieve the recommendations made by Emergency Nursing Association (ENA), which state that a replacement nurse should get on a 24 h long supervision practice for triage [31]. Furthermore, consistent with Faheim et al. [5], poor performance on triage practice could be due to deficient knowledge, absence of orientation for newly graduated and newly recruited nurses, insufficient materials and equipment so rotation to emergency care departments which are better resourced could enhance the knowledge and practice of emergency care and triage.

### 6.3. Sub-Theme 2.3: Importance of Supervisors Allocated in the Triage Area Outlined

Emergency nurses indicated that triage practice could be enhanced by an available clinical supervisor, especially to the newly employed nurses that need encouragement. Furthermore, clinical supervisors can improve triage practices by giving triage feedback, especially on the identified gaps. The following quotes support the findings:Participant 13 (<6 years of working experience): “*Supervision of newly employed nurses, encouraging nurses to do in-service (sometimes in your presence) so that you can identify the gaps, put complete triage form on triage board or wall so that it will aid them on how to complete triage form and also encouraging subordinates to ask questions where they don’t understand*.”Participant 11 (<6 years of working experience: “*There should be a supervisor who always checks triage forms so that the department improves from the loopholes*.”Participant 5 (<6 years of working experience: “*…having a leader/supervisor give feedback that will enhance the triage practice from the feedback*.”Participant 16 (>10 years of working experience: “*Shift leader to give feedback every day re-performance of staff and corrections where needed*.”

Clinical supervision plays a critical role in enhancing the medical resident adherence to guidelines in the ED, and these effects of clinical supervision of health professionals on patient safety have been established in recent systematic reviews. The reviews investigated the effects of experienced health professionals guiding the practice of less experienced professionals, the authors concluded that clinical supervision is associated with a reduced risk of adverse patient outcomes [35]. According to Hategeka, Mwai, and Tuyisenge [29], Complementing the Emergency Triage Assessment Treatment (ETAT) training with regular supervision and mentorship could help ensure that knowledge translation takes place and identify further opportunities to enhance the impact of the ETAT program.

According to Yuliandari [36], several studies have recommended a strategy of clinical supervision to enhance novice clinical skills in the ED; experienced nurses need to educate novice nurses by mentoring and escorting when making clinical decisions, and this will boost their confidence and competence. Whenever possible, having a reflective session after a decision has been taken is advantageous to prevent biases and justify and clarify a decision. The reflective session may again equip novice triage nurses with practical skills of assessing patients in the triage room by reflecting upon their actions, based upon the assumption that the expert nurses’ abilities in triage decision making could be transferable to the novice nurse.

## 7. Limitations of the Study

The study was conducted on the selected five hospitals of Sekhukhune District only and cannot be generalized to other hospitals outside the district and the province. The study excluded the knowledge of nurse managers working in the ED in developing strategies to enhance triage. The focus of the study was only on professional nurses working in ED, and therefore, further studies might be implemented to target the nurse managers on the same phenomenon.

## 8. Recommendation

The study findings indicated a description of measures to enhance practices amongst emergency unit staff. Professional nurses reported a formulation of clear guidelines for handling patients in the triage area and benchmarking on managing the triage area as possible strategies to enhance triage practice. Emergency nurses further highlighted that triage practice could be enhanced by clinical supervisors’ availability in the triage areas who could enhance practice by providing feedback on the identified gaps, especially to the newly employed nurses who need encouragement.

## 9. Conclusions

The study, concludes that there are various strategies to enhance the knowledge and practice of triage amongst nurses in ED as described by emergency nurses. Emergency nurses highlighted various platforms that can enhance the knowledge and practice of triage. These include continuous training, provision of triage courses, evaluating the developed guidelines for triage implementation, and benchmarks between hospitals having ED. We recommend further studies that may implement and evaluate the effectiveness of the discussed strategies in enhancing nurses’ knowledge and practical skills within the ED of rural hospitals in SA.

## Figures and Tables

**Table 1 ijerph-18-04471-t001:** Sekhukhune District Hospitals.

Hospitals	Municipal Type
Dilokong Hospital	Greater Tubatse Municipality
Mecklenburg	Greater Tubatse Municipality
Jane Furse Hospital	Makhuduthamaga Municipality
Groblersdal Hospital	Elias Motswaledi Municipality
Matlala Hospital	Greater Marble Hall Municipality

**Table 2 ijerph-18-04471-t002:** Demographic data of participants (*N* = 17).

Variables	Totals (%)
Gender(M/F)	
Female	9(53%)
Male	8(47%)
Qualification type	
General professional nurse	9 (53%)
M.Cur (Masters in Nursing), Trauma speciality	1 (6%)
Diploma in Nursing (Trauma Specialty)	7(41%)
Working Experience	
Less than 5 years	6 (35%)
6–10 years	4 (24%)
Above 10 years	7 (41%)

**Table 3 ijerph-18-04471-t003:** Summary of the main findings according to themes and sub-themes and categories.

Themes	Sub-Themes	Categories
1. The consistent description of the importance of triage training for emergency unit staff	1.1 Participants emphasized the upskilling of emergency nurses on triage as an important enhancement factor of the existing triage knowledge through various training platforms 1.2 Triage course/module as an orientation course and guidelines for implementation suggested	Upscaling of emergency skills Orientation course
2. The description of measures to enhance triage practices among emergency unity staff	2.1. Formulation of clear guidelines for handling patients in the triage area viewed as important 2.2. Benchmarking on the management of the triage area to improve practice suggested 2.3. Importance of supervisors allocated in the triage area outlined	Clear guidelines Benchmarking Supervision

## Data Availability

Data generated and analyzed during the current study are not publicly available due to ethical reasons but are available for corresponding authors.

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
