# Peer review of "Strategies to Enhance Knowledge and Practical Skills of Triage amongst Nurses Working in the Emergency Departments of Rural Hospitals in South Africa"

_ijerph, 2021, doi:10.3390/ijerph18094471_

Round 1

Reviewer 1 Report

For your evaluation, we suggest that you can increase your recommendations conclusions, based on your results.

Example:

Incorporate in Conclusion recommendation:

aspects investigated and that appear in Results and Discussion such as: line (330) Sub-theme 2.3: Importance of supervisors allocated in the triage area outlined, or also, line (239) recommend over 6. THEME 2: Description of measures to enhance practices amongst emergency unit staff

Reviewer 2 Report

" Please See the attachment "

Reviewer 3 Report

Specify the type of qualitative sampling used. It is indeed non-probabilistic, as it is a qualitative research, but you must specify what type of qualitative sampling has been used to set up the sample.

Introduction:

In general, very well structured. Well based on the bibliography and correct justification of the need for the topic of study.

Page 2, line 45: "For instance, [6] reported...", include as subject of the sentence the surname of the author of this research. The same should be done in line 47 of the same page: "Similarly, a study by [7] reported...".

Methods:

Page 2, lines 88-91. Justification of the need for study with a qualitative approach should be made at the end of the introduction, not in this section of methods.

In this first paragraph of Methods, the objectives of the research should be further detailed.

In the methods section, in the text, reference should be made to the point at which table 1 should be displayed. It would be interesting to include in this table the distribution of participating nurses by the different hospitals included.

The fact that the data collection process took place in the hospital itself during work activity is a limitation of this research. The participants could not dedicate enough time or provide the necessary information to fully cover the objectives of the research, as they had care activities. This is evident from the short duration of the interviews carried out, which may not have achieved adequate saturation of the information obtained.

Page 3, line 123-124, include subject in the sentence associated with the reference [22].

Explain whether it was the same person who conducted the interviews that transcribed them or someone else.

Provide more information on the interview guide used.

Was it necessary to conduct more than one interview session with any participant?

The analysis process by which the themes and sub-themes were generated should be explained in more detail.

Was feedback given to the participants on the topics obtained in the analysis?

Results:

Do not repeat in the text (page 4, lines 133-143) the information already collected in table 2.

The labels used to name the themes and sub-themes are very extensive. An effort should be made to synthesize them, while at the same time they should be of a higher level of abstraction or conceptualization. In general, the analysis is shallow and very much linked to the data.

The quotations of the participants must be accompanied by some characteristic of the participant profile that expresses this quotation (for example, sex and professional seniority or age).

Review throughout the results discussion, that all sentences have a subject when a bibliographic reference is added.

The fact that the results cannot be extrapolated because they are a qualitative approach is not a limitation. Qualitative studies do not have this objective and it is a mistake to say that this is a limitation of this study. One limitation would be not having also interviewed hospital managers to know their vision on this problem and to study their training strategies and how to implement them.

It is advisable to use a quality guide for qualitative studies (e.g. COREQ), during the development and reporting of the research.

Round 2

Reviewer 2 Report

According to the comments on the manuscripts review, it was confirmed that each item was modified with an upgraded content.
Thank you.

Author Response

Comments believe they have responded to the reviewers comments on reviewers report one.

Reviewer 3 Report

Once I have review your paper, again, I need to do the follow considerations. Although several changes have been done, the authors have not answered to the follow recommendations o asks previous:
- Was feedback given to the participants on the topics obtained in the analysis?

- Review throughout the results discussion, that all sentences have a subject when a bibliographic reference is added.

- The fact that the results cannot be extrapolated because they are a qualitative approach is not a limitation. Qualitative studies do not have this objective and it is a mistake to say that this is a limitation of this study. One limitation would be not having also interviewed hospital managers to know their vision on this problem and to study their training strategies and how to implement them.

- It is advisable to use a quality guide for qualitative studies (e.g. COREQ), during the development and reporting of the research.
